# Dual Targeting of EZH2 Degradation and EGFR/HER2 Inhibition for Enhanced Efficacy against Burkitt’s Lymphoma

**DOI:** 10.3390/cancers15184472

**Published:** 2023-09-08

**Authors:** Se Been Kim, Chae-Eun Yang, Yurim Jeong, Minseo Yu, Wan-Su Choi, Jung-Yeon Lim, Youngwoo Jeon

**Affiliations:** 1Department of Biomedical Laboratory Science, Inje University, Gimhae 50834, Republic of Korea; kimecc@naver.com (S.B.K.); sophiss@naver.com (C.-E.Y.); jyrim0301@naver.com (Y.J.); yms02625@naver.com (M.Y.); wansuchoi@inje.ac.kr (W.-S.C.); 2Lymphoma and Cell-Therapy Research Center, Yeouido St. Mary Hospital, School of Medicine, The Catholic University of Korea, Seoul 07345, Republic of Korea; 3JL’s Lymphoma Origins & Clinical Applications Lab (JL-LOCAL), The Catholic University of Korea, Seoul 07345, Republic of Korea; 4Department of Digital Anti-Aging Health Care, Inje University, Gimhae 50834, Republic of Korea; 5Department of Hematology, Yeouido St. Mary Hospital, School of Medicine, The Catholic University of Korea, Seoul 07345, Republic of Korea

**Keywords:** lymphoma, EZH2, HER2/neu-EGFR, PROTAC, apoptosis

## Abstract

**Simple Summary:**

Burkitt’s lymphoma (BL) is an aggressive type of non-Hodgkin lymphoma that originates from B-cells. It is characterized by translocation of the MYC gene, leading to the upregulation of this gene and subsequently causing overexpression of EZH2. Since BL shows rapid involvement, the development of a new effective agent is needed. In this study, we targeted EZH2 and HER2/neu-EGFR, both of which are related to cell proliferation and overexpression-induced tumors. We conducted a study using a combination of MS1943, EZH2 degrader, lapatinib, and HER2/neu-EGFR inhibitor. We demonstrated that a combination of MS1943 and lapatinib induced apoptosis in Daudi cells with S and G2/M phase arrest in Ramos and Daudi cells. These promising agents could prove to be a new therapeutic option against BL.

**Abstract:**

EZH2, a histone methyltransferase, contributes significantly to cancer cell survival and proliferation. Although various EZH2 inhibitors have demonstrated promise in treating lymphoma, they have not fully managed to curb lymphoma cell proliferation despite effective reduction of the H3K27me3 mark. We used MS1943, an EZH2 selective degrader, which successfully diminishes EZH2 levels in lymphoma cells. Additionally, lapatinib, a dual inhibitor of the epidermal growth factor receptor (EGFR) and human epidermal growth factor receptor 2 (HER2) tyrosine kinases, targets a receptor protein that regulates cell growth and division. The overexpression of this protein is often observed in lymphoma cells. Our study aims to combine these two therapeutic targets to stimulate apoptosis pathways and potentially suppress Burkitt’s lymphoma cell survival and proliferation in a complementary and synergistic manner. We observed that a combination of MS1943 and lapatinib induced apoptosis in Daudi cells and caused cell cycle arrest at the S and G2/M phases in both Ramos and Daudi cells. This strategy, using a combination of MS1943 and lapatinib, presents a promising therapeutic approach for treating lymphoma and potentially Burkitt’s lymphoma.

## 1. Introduction

Burkitt’s lymphoma (BL) is a highly aggressive B-cell lymphoma characterized by a rapid proliferation rate and translocation of the MYC gene [1,2]. BL shows features of male dominance [3] and extranodal involvement in the central nervous system (CNS) in particular [4,5] and has poor prognostic factors, including a high level of LDH and advanced stage at diagnosis [6,7]. In the case of receiving aggressive chemotherapy, pediatric and adult patients showed 90% and 5~60% of disease-free status, respectively [8]. It is hard to proceed with intensive chemotherapy in the elderly, and patients who have CNS involvement are at high risk of the failure of this treatment [8], necessitating the development of innovative therapeutic strategies.

EZH2 is a critical component of the polycomb repressive complex 2 (PRC2) and plays a vital role in the regulation of epigenetic gene expression. Aberrant EZH2 activity, often caused by gain-of-function mutations, has been implicated in the pathogenesis of BL [9]. The dysregulation of EZH2 leads to the repression of tumor suppressor genes, promoting uncontrolled cell growth and survival in BL. Targeting EZH2 presents a potential therapeutic strategy to restore normal epigenetic regulation and impede the progression of this disease. Various EZH2 inhibitors, including UNC1999, C24, EPZ6438, GSK126, CPI-12051, and PF-06821497, have been developed to inhibit EZH2/PRC2 activity and reduce levels of H3K27me3 [10,11]. As H3K27me3 is involved in target gene repression and can generally suppress oncogenesis [11], these inhibitors have shown promise in clinical development for treating various tumors including sarcomas, lymphomas, and MRT. The inhibition of EZH2/PRC2 enzymatic activity effectively blocks tumor cell growth, metastasis, and angiogenesis of tumor tissue [11]. Despite the development of EZH2 inhibitors, their efficacy has been limited [10].

To overcome this challenge, alternative strategies such as PROTACs (proteolysis targeting chimeras) and hydrophobic tagging have emerged as successful approaches for the selective degradation of target proteins [12,13]. MS1943, an EZH2 selective degrader, effectively reduces EZH2 levels in breast cancer [12,13]. These innovative technologies may offer a novel route for enhancing the therapeutic potential of EZH2 inhibition for the treatment of BL.

The HER2/neu-EGFR signaling pathway plays a crucial role in normal B-cell development and function. The EGFR/HER2 pathway triggers c-MYC activation via cytosolic phosphorylation and downstream RAS-RAF-MEK-ERK signaling. This pathway’s overexpression can drive tumorigenesis. A dual approach—combining an EZH2-degrading drug and a HER2 inhibitor—to counteract the effects of c-Myc-induced EZH2 upregulation and the HER2-initiated pathway has been investigated [14]. HER2 and EBV play significant roles in human gastric cancer by driving cancer progression and metastasis through epithelial–mesenchymal transition (EMT) [15]. This study explores a potential link between HER2 and EBV oncoproteins in gastric cancer, where EBV triggers EMT via PI3K/Akt and Shc-MAPK/Erk1/Erk2 pathways, while HER2 activates EMT-related pathways through phosphorylation, contributing to cancer advancement. Leveraging the relevance of this pathway, we conducted drug combination experiments involving an HER2 inhibitor and EZH2 degrader in our Burkitt’s lymphoma cell line with EBV oncoprotein [15]. Abnormal activation of this pathway, resulting from gene amplification, overexpression, or activating mutations, has been observed in various solid tumors, including breast cancer, thyroid carcinoma, and gastric carcinoma [16,17,18]. Dysregulated HER2/neu-EGFR signaling contributes to cell survival, proliferation, and apoptosis resistance, underlining its therapeutic target potential [19,20]. Lapatinib, a dual inhibitor of EGFR and HER2 tyrosine kinases, has demonstrated potential in breast cancer treatment and other solid tumors in clinical trials [21,22]. The targeted inhibition of this pathway holds potential for developing effective therapeutic strategies in BL.

In conclusion, our findings indicate that the combination of MS1943 and lapatinib has a synergistic effect, promoting apoptosis and enhancing anti-tumor activity in BL. These results suggest that this synergistic approach holds promise as a therapeutic strategy for BL treatment.

## 2. Materials and Methods

### 2.1. Cell Lines and Cell Culture

In this study, the BL cell lines Ramos (RA 1) (ATCC CRL-1596) and Daudi (ATCC CCL-213) were used. Both cell lines were obtained from the Korean Cell Line Bank (Seoul, Republic of Korea). These cell lines were grown in RPMI 1640 medium (Gibco, Carlsbad, CA, USA) supplemented with 10% heat-inactivated fetal bovine serum (FBS; Gibco), 2 mM L-glutamine (Gibco), and 1% antibiotics (10 U/mL penicillin and 10 g/mL streptomycin; Gibco). All cell lines were incubated at 37 °C and 5% CO_2_.

### 2.2. Drug Preparations

Lapatinib (HER2/neu-EGFR inhibitor) and MS1943 (EZH2 degrader) were purchased from MedChemExpress (Monmouth Junction, NJ, USA). Both were dissolved in dimethyl sulfoxide (DMSO) as recommended by the manufacturer and stored at −80 °C. All these drugs were diluted with the RPMI 1640 medium with 10% heat-inactivated FBS, 2 mM L-glutamine, and 1% antibiotics before being used for the treatment of the cell lines.

### 2.3. Cell Counting Kit-8 (CCK-8) Assay

Cell growth was assessed using the CCK-8 (Dojindo, Rockville, MD, USA) assay according to the manufacturer’s protocol. The Ramos and Daudi cell lines were seeded at an initial cell density of 2 × 10^4^ cells/100 μL culture medium in 96-well plates. These cells were treated with various doses of lapatinib and MS1943 drugs without any drugs, and with 2.5 μM, 5 μM, and 10 μM for each drug. The synergistic effects were assessed using 5 μM for both drugs. The cultures were maintained at 37 °C in a 5% CO_2_ atmosphere. CCK-8 solution was added in increments of 10 μL to each well after 24 h, 48 h, and 72 h. The plates were incubated for 1–4 h in a CO_2_ incubator and the optical density was measured at 450 nm using a microplate reader.

### 2.4. Western Blotting

One million Ramos or Daudi cells were lysed in 2× laemmli sample buffer (Bio-Rad) with β-mercaptoethanol and boiled at 95 °C for 10 min. After removal of the insoluble fraction by centrifugation at 10,000× *g* for 10 min, protein samples were separated by SDS gel electrophoresis and transferred to a polyvinylidene difluoride membrane. The membranes were stained with cleaved PARP, cleaved caspase-3, and PUMA antibodies (Cell Signaling Technology, Danvers, MA, USA) at a dilution of 1:1000 or GAPDH antibody (Cell Signaling Technology) at a dilution of 1:2000 at 4 °C overnight. After overnight incubation at 4 °C, HRP-conjugated secondary antibodies were added. After washing with Tris-buffered saline and Tween 20, the hybridized bands were detected using an enhanced chemiluminescence (ECL) detection kit (Amersham Pharmacia Biotech, Buckinghamshire, UK).

### 2.5. RNA Extraction and cDNA Synthesis

The 5 × 10^5^ cells were treated with or without drugs as described previously. Total RNA was extracted using TRIzol reagent (Invitrogen, Waltham, MA, USA) and the concentration of RNA was measured using the NanoDrop. These were reverse transcribed using a High-Capacity RNA-to-cDNA Kit (Appliedbiosystems, Waltham, MA, USA) following the manufacturer’s protocol. The cDNA was synthesized using a Thermocycler (Bio-Rad, Hercules, CA, USA) with cycling conditions including primer annealing at 25 °C for 10 min, DNA polymerization at 37 °C for 120 min, and finally reverse transcriptase deactivation at 85 °C for 5 min. The synthesized cDNA was stored at −20 °C before further use.

### 2.6. Real-Time Reverse Transcription PCR (RT-PCR)

RT-PCR was performed using the TB Green^®^ Fast qPCR Mix (TaKaRa Bio, Otsu, Japan) using the manufacturer’s protocol as follows: hold 1 cycle for 30 s at 95 °C, 2 step PCR 40 cycles for 5 s at 95 °C and 30 s at 60 °C. Dissociation steps consisted of 15 s at 95 °C, 30 s at 60 °C, and 15 s at 95 °C for 1 cycle. For detection of the mechanisms of lapatinib and MS1943, the following gene-specific primers were used: Chop (forward: 5′-GCACCTCCCAGAGCCCTCACTCTCC-3′; reverse: 5′-GTCTACTCCAAGCCTTCCCCCTGCG-3′), Bip (forward: 5′-CGAGGAGGAGGACAAGAAGG-3′; reverse: 5′-CACCTTGAACGGCAAGAACT-3′), Bax (forward: 5′-CTGCAGAGGATGATTGCCG-3′; reverse: 5′-TGCCACTCGGAAAAAGACCT-3′), p53 (forward: 5′-GTTCCGAGAGCTGAATGAGG-3′; reverse: 5′-TCTGAGTCAGGCCCTTCTGT-3′), cyclin D1 (sense: ATGCCAACCTCCTCAACGAC; antisense: GGCTCTTTTTCACGGGCTCC), Xbp1 (forward: 5′-TGCTGAGTCCGCAGCAGGTG-3′; reverse: 5′-GCTGGCAGGCTCTGGGGAAG-3′), β-actin (forward: 5′-TCACCCACACTGTGCCCATCTACGA-3′; reverse: 5′-AGCGGAACCGCTCATTGCCAATG-3′), and GAPDH (forward: 5′-CCACTCCTCCACCTTTGACG-3′; reverse: 5′-CCACCACCCTGTTGCTGTAG-3′). For quantification, the relative mRNA expression of specific genes was calculated using the 2^−ΔΔCt^ method after normalization to GAPDH expression.

### 2.7. Cell Cycle Analysis

The 5 × 10^5^ cells were treated with the drugs described. After incubating for 72 h in a CO_2_ incubator, the cells were harvested and centrifuged at 2000 rpm for 5 min with DPBS (Gibco) 2 mL. These cells were then washed twice and the supernatant discarded. For cell cycle analysis, the Annexin V-PI Apoptosis Kit (BioVision, London, UK) was used following the manufacturer’s protocol. The cell cycle was detected using the BD LSRFortessa (Taunton, MA, USA).

### 2.8. Statistical Analysis

Statistical significance was determined using Student’s two-tailed *t*-test and one-way analysis of variance (ANOVA) with Bonferroni correction for multiple comparisons. Differences between arthritis incidences at a given time point were evaluated by χ2 contingency analysis. In all analyses, *p* values less than 0.05 were considered to indicate statistical significance.

## 3. Results

### 3.1. MS1943 (EZH2 Degrader) and Lapatinib (EGFR/HER2 Inhibitor) Inhibit Cell Growth in BL Cells

The CCK-8 assay was used to investigate the effect of treatment with MS1943 and lapatinib on the proliferation of Ramos and Daudi cells. Both the Ramos and Daudi cells were treated with different concentrations of MS1943 and lapatinib (DMSO—2.5 μM, 5 μM, and 10 μM) for 24, 48, and 72 h (Figure 1A–H). These results demonstrated that the MS1943 and lapatinib inhibited proliferation of the BL cells.

### 3.2. MS1943 Synergizes with Lapatinib Drugs in BL Cell Lines, Resulting in Significant BL Proliferation Inhibition

To evaluate the suppressive effects of MS1943 and lapatinib on Ramos and Daudi cell proliferation in vitro, we cultured cells in media treated with or without MS1943 and lapatinib with CCK-8 assay analysis. As expected, the combination of MS1943 and lapatinib resulted in a significant reduction of the Ramos and Daudi cells compared with single-drug treatments (Figure 2A,B). These results suggest that a combined approach could effectively inhibit BL cell survival and proliferation, demonstrating synergistic effects.

### 3.3. Combination of MS1943 and Lapatinib Therapy Induces Prolonged Activation of the UPR Pathway in BL Cells

Adaptation to protein-folding stress involves the activation of distinct stress sensors located at the endoplasmic reticulum (ER) membrane. One well-established arm of the unfolded protein response (UPR) is mediated by IRE1α. Upon activation, IRE1α utilizes its endoribonuclease activity to splice Xbp1 mRNA, leading to the production of functional XBP1. XBP1 acts as a transcription factor, regulating specific sets of target genes in a cell type-specific manner [23]. In our study, we employed qPCR to confirm the upregulation of Xbp1 and its downstream effectors Chop and Bip in Ramos and Daudi cells following treatment with MS1943 and lapatinib for 72 h. We observed that the increased expression of Bip, Chop, and Xbp1 in BL led to apoptosis. However, upregulation of the UPR-related gene Xbp1 showed a relatively different pattern in the Ramos cell line, with the control showing the highest mRNA expression level. In response to these results, UPR occurred more significantly in the Daudi cell line. These findings provide evidence of the induction of UPR signaling and activation of XBP1-mediated gene regulation in response to combination treatment (Figure 3A–C) [13].

### 3.4. Combination of MS1943 and Lapatinib Leads to BL Cell Cycle Arrest

Cell cycle analysis was conducted to investigate the mechanisms of the blocking of EZH2 and EGFR/HER2 signaling for the proliferation of BL cells. Cell cycle distribution was altered by the inhibition of both EZH2 and HER2/neu-EGFR signaling. The proportion of G2/M phase cells was decreased and the proportion of S phase cells was induced by down-regulated EZH2 and HER2/neu-EGFR signaling in the Ramos (Figure 4A–C) and Daudi cell lines (Figure 4D,E). We performed a qPCR assay on the MS1943- and lapatinib-treated BL cell lines. The expression of the cell cycle arrest gene Cyclin D1 level was significantly increased in the combinatorial drug-treated cells compared to the single drug in the Daudi cells (Figure 4F). Based on these data findings, we hypothesized that the downregulation of both EZH2 and EGFR/HER2 signaling may inhibit proliferation by inducing cell cycle arrest in Daudi cells.

### 3.5. Synergistic Effects of MS1943 and Lapatinib Promote Apoptotic Cell Death

To examine the apoptotic effects of Ramos and Daudi cells, MS1943 were treated with or without lapatinib for 72  h and then stained with annexin V and propidium iodide (PI) [24]. As is shown in Figure 5A–F, a significant increase in the apoptosis and necrosis rates of the combination group in both the Ramos and Daudi cell lines were observed (Figure 5A–E). On the other hand, it was confirmed that live cells were significantly reduced in the combination group (Figure 5A,D). The ability of the combination group to promote apoptosis was significantly stronger in the Daudi cells compared to the Ramos cells (Figure 5A–C).

### 3.6. Combination of MS1943 and Lapatinib Induces Apoptosis through Caspase Pathway

In order to carry out apoptosis, cells must activate the caspase family [25,26]. To verify whether blocking of both EZH2 and HER2/neu-EGFR induces the activation of these caspases, Ramos and Daudi cells were exposed to MS1943 and lapatinib drugs. The protein levels and cleaved forms of caspases-3, PARP, and PUMA were examined by Western blot analysis (Appendix A). Blocking of EZH2 and EGFR/HER2 signaling induced the activation of caspase-3, which is one of the characteristics of apoptosis, with the activation of PUMA levels and cleavage of the PARP protein into 116 kDa (pro-form) and 89 kDa (cleaved form) fragments through active caspase-3 (Figure 6A–D,F–I). Previous studies have demonstrated that p53 phosphorylation induces pro-apoptotic genes such as the Bcl-2-associated X protein (Bax) and the p53-upregulated modulator of apoptosis (PUMA) [27]. Our data indicated that the MS1943- and lapatinib-treated Daudi cells significantly induced p53 (Figure 6J) and Bax expression (Appendix A) in the Daudi cells, resulting in lymphoma cell apoptosis. However, the MS1943- and lapatinib-treated Daudi cells exhibited no difference in p53 levels in the qPCR (Figure 6E).

## 4. Discussion

In a clinical setting, EZH2 has emerged as a prominent factor in various lymphomas [9,28]. However, despite initial expectations, EZH2 inhibitors have yet to yield favorable outcomes in patients [29,30]. Our study aimed to address this discrepancy by evaluating the comparative efficacy of tazemetostat, an FDA-approved EZH2 inhibitor, and MS1943, an EZH2 degrader, across diverse lymphoma cell lines.

Intriguingly, our findings revealed that MS1943 exerted a significantly more potent inhibitory effect on cell proliferation compared to tazemetostat (Appendix A). This observation highlights the potential of utilizing PROTAC-based EZH2 degradation as a therapeutic strategy for lymphoma treatment. Furthermore, our study sought to explore the synergistic effects of combining MS1943 with lapatinib, a well-established drug targeting HER2/neu-EGFR signaling, for the treatment of BL.

Our results revealed that treatment with MS1943 and lapatinib exhibited a dose-dependent anti-proliferative effect on BL cell lines such as Ramos and Daudi cells (Figure 1). Importantly, the combination of MS1943 and lapatinib exhibited synergistic effects, leading to a significant reduction in cell proliferation compared to single-drug treatment (Figure 2). These results demonstrated that the expression of the cell cycle arrest gene Cyclin D1 was significantly increased in the combination treatment, suggesting a disruption in cell cycle progression. Further analysis revealed that combination therapy induced apoptosis in Daudi cells (Figure 6). Moreover, the combination of MS1943 and lapatinib led to the induction of Bax expression (Appendix A), key regulators of apoptotic pathways. These findings support the notion that the combined targeting of EZH2 and HER2/neu-EGFR pathways can effectively promote apoptosis in BL cells. However, it is important to note that our results revealed slight differences in the response of Ramos and Daudi cells. Daudi cells are known to be Epstein–Barr virus (EBV) positive, while Ramos cells are EBV negative. This distinction suggests the need for additional investigations to elucidate the underlying mechanisms and analyze the impact of EBV status on the observed outcomes. Different cellular characteristics and signaling pathways influenced by EBV infections may contribute to varying treatment responses, warranting a more in-depth analysis to unravel the intricacies of these distinct cellular contexts. Future studies should focus on dissecting the specific mechanisms involved to better understand the differential response of these cell lines and optimize treatment strategies accordingly.

Additionally, our data demonstrated that combination therapy resulted in the induction of prolonged endoplasmic reticulum (ER) stress, as evidenced by the upregulation of UPR-associated genes including Xbp1, Chop, and Bip (Figure 3). This indicates that ER stress may contribute to the apoptotic effects observed in BL cells treated with MS1943 and lapatinib.

Finally, the activation of caspase-3, along with the cleavage of PARP and activation of PUMA, further supported the induction of apoptosis by combination therapy. These findings highlight the involvement of the caspase pathway in the apoptotic process induced by blocking both EZH2 and HER2/neu-EGFR signaling (Figure 6).

## 5. Conclusions

Our study provides compelling evidence for the potential of a combined therapeutic strategy utilizing MS1943 and lapatinib in BL. The synergistic effects observed in our experiments, including enhanced anti-proliferative activity, induction of apoptosis, cell cycle arrest, and activation of the caspase pathway, highlight the promise of this approach in improving clinical outcomes for patients with BL. Future investigations and clinical trials are warranted to validate the efficacy and safety of this combination therapy in a clinical setting.

## Figures and Tables

**Figure 1 cancers-15-04472-f001:**
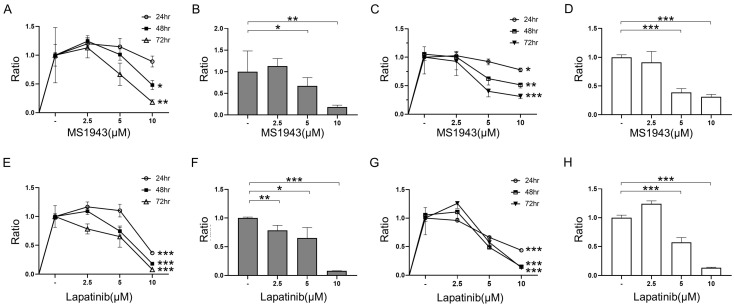
Cell viability after treatment with MS1943 and lapatinib was measured using the CCK-8 assay. (**A**) The Ramos cell line was exposed to 2.5 μM, 5 μM, and 10 μM of MS1943 and observed for 24, 48, and 72 h. (**B**) Meaningful results were shown in 72 h with 2.5 μM, 5 μM, and 10 μM of treatment. (**C**) In the Daudi cell line, MS1943 was treated with different concentrations as described above for 24, 48, and 72 h. (**D**) Treatment of MS1943 in the Daudi cell line exhibited dose-dependent inhibition for 72 h. (**E**) Ramos cells were exposed to 2.5 μM, 5 μM, and 10 μM of lapatinib and observed for 24, 48, and 72 h. (**F**) Ramos cell viability indicated dose-dependent inhibition in 72 h. (**G**) In the Daudi cell line, lapatinib was treated with different concentrations as described above for 24, 48, and 72 h. (**H**) Treatment of lapatinib in the Daudi cell line led to dose-dependent inhibition for 72 h. *, *p* < 0.05; **, *p* < 0.01; ***, *p* < 0.001 as determined by two-tailed, unpaired *t*-tests. Error bars are shown as mean ± SD.

**Figure 2 cancers-15-04472-f002:**
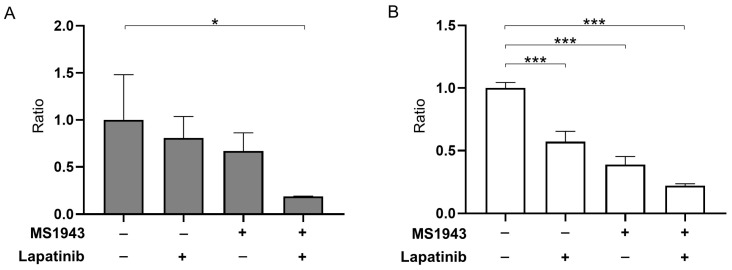
Drug combination therapy for 72 h showed synergistic effects in the CCK-8 assay. (**A**) The Ramos cell line was exposed to 5 μM of MS1943 and lapatinib each or in combination. Combination treatment exhibited a significant reduction compared with the DMSO control and single-treated drug. (**B**) Daudi cell viability was assayed using CCK-8 analysis and the represented co-treatment demonstrated promising effects against the control and single therapy. *, *p* < 0.05; ***, *p* < 0.001 as determined by two-tailed, unpaired *t*-tests. Error bars are shown as mean ± SD.

**Figure 3 cancers-15-04472-f003:**
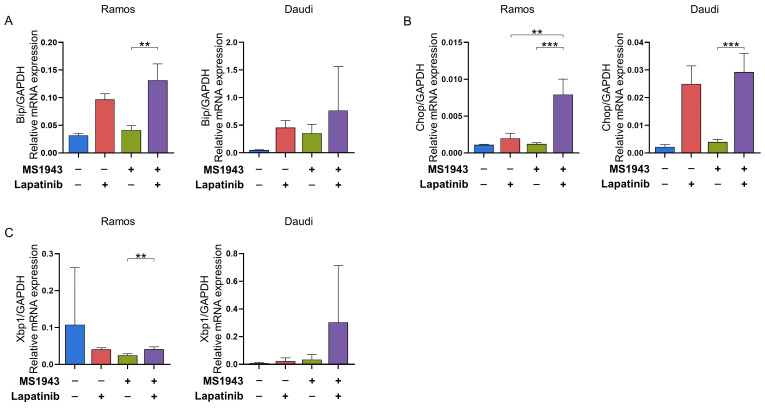
Drug combination induced UPR pathway among all related genes by qRT-PCR. (**A**–**C**) Increased expression of UPR-related genes such as Bip, Chop, and Xbp1 caused the accumulation of unfolded proteins and ER stress, leading to apoptosis in the Ramos and Daudi cell lines. UPR occurred more significantly in the Daudi cell lines. **, *p* < 0.01; ***, *p* < 0.001; as determined by two-tailed, unpaired *t*-tests. Error bars are shown as mean ± SD.

**Figure 4 cancers-15-04472-f004:**
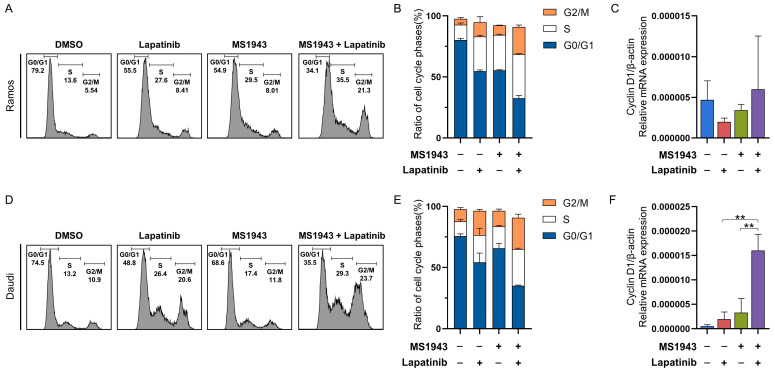
Cell cycle analysis revealed apoptosis by arresting S phases. (**A**) Ramos cells were treated with control, MS1943, and lapatinib for each or in combination for 72 h and detected by flow cytometry. (**B**) The relative ratio of S phases expanded in both MS1943 and lapatinib treatment. S phase-arrest indicates apoptosis. (**C**) The synergistic effects of MS1943 and lapatinib induced cell cycle arrest through Cyclin D1 upregulation in Ramos cells. (**D**) Daudi cells were exposed in the same fashion as the Ramos cells. (**E**) The relative ratio of S phases largely increased compared to the control and single-treated cells, representing cell death by apoptosis. (**F**) The synergistic effects of MS1943 and lapatinib induces cell cycle arrest through Cyclin D1 upregulation in lymphoma. The experiment was repeated in duplicate and merged data from all the experiments are shown. **, *p* < 0.01; as determined by two-tailed, unpaired *t*-tests. Error bars are shown as mean ± SD.

**Figure 5 cancers-15-04472-f005:**
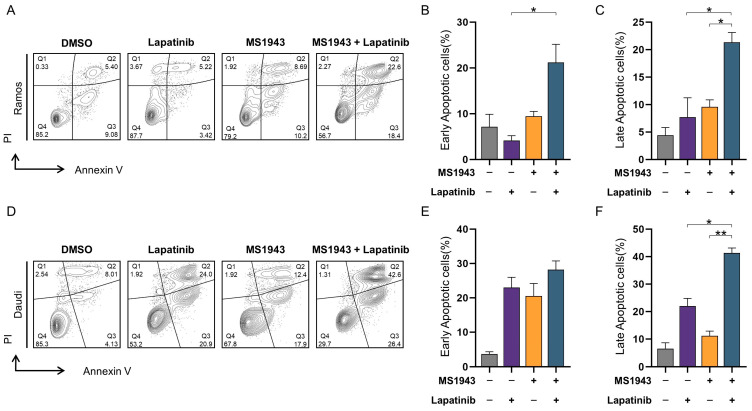
Apoptotic effects of combination treatment on Ramos and Daudi cell were examined by Annexin V staining. Each section indicates Necrosis cells (Q1), Late Apoptotic cells (Q2), Early Apoptotic cells (Q3) and Live cells (Q4). (**A**) Ramos cells were treated with 5 µM of control, lapatinib, MS1943, and a combination thereof. Apoptosis was evaluated after 72 h of treatment. (**B**,**C**) The relative percentage of Early Apoptotic cells (Q3) and Late Apoptotic cells (Q2) were meaningfully increased compared with the control. (**D**) Daudi cells were treated in the same way as Ramos cells. (**E**,**F**) The relative percentages of Early Apoptotic cells (Q3) and Late Apoptotic cells (Q2) were significantly increased against the control. These showed that dual-targeting therapeutic combinations induced apoptosis effectively. The experiment was repeated in duplicate and merged data from all the experiments are shown. *, *p* < 0.05; **, *p* < 0.01; as determined by two-tailed, unpaired *t*-tests. Error bars are shown as mean ± SD.

**Figure 6 cancers-15-04472-f006:**
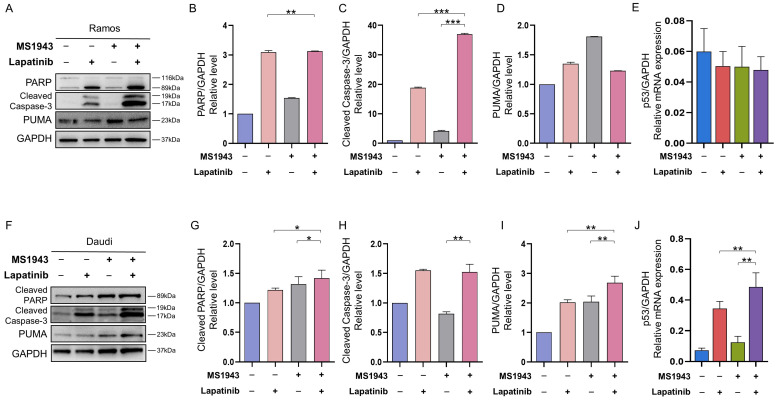
Drug combination induced upregulation of apoptosis-associated proteins. (**A**) The Ramos samples were exposed to 5 μM MS1943 and lapatinib each or in combination for 72 h. (**B**–**D**). Western blot results were analyzed relatively by using GAPDH. All the proteins associated with apoptosis were highly increased in both MS1943- and lapatinib-treated cells. (**E**) Combination treatment in the Daudi cells exhibited distinctive outcomes with increased relative mRNA expression of p53. (**F**) Daudi cells were exposed to the same 5 μM as mentioned above for 72 h. (**G**–**I**) Western blot results were analyzed relatively and indicated upregulation in the combination therapeutic agents. This demonstrates that the drug combination induced apoptosis effectively in protein levels. (**J**) Combination treatment in the Daudi cells showed distinctive outcomes with increased relative mRNA expression of p53. Western blot total gel can be found in Appendix A. *, *p* < 0.05; **, *p* < 0.01; ***, *p* < 0.001 as determined by two-tailed, unpaired *t*-tests. Error bars are shown as mean ± SD.

## Data Availability

Data sharing is not applicable to this article.

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
