# Peer review of "Dual Targeting of EZH2 Degradation and EGFR/HER2 Inhibition for Enhanced Efficacy against Burkitt’s Lymphoma"

_cancers, 2023, doi:10.3390/cancers15184472_

Round 1
Reviewer 1 Report
The authors test EZH2 degradation and EGFR2/Her2 inhibition on 2 Burkitt lymphoma cell lines (Daudi and Ramos). The technical presentation is in order, but I have some remarks with the background and rational of the study.
1. In the simple summary it says translocation of the gene: which gene? And mainly associated with EBV: there is quite a difference in endemic (EBV+) and sporadic (18-40% EBV+) BL, in this case it is probably sporadic, so the association with EBV is less. The most important factor is probably the upregulation of MYC which induces overexpression of EZH2, and is more important fort his work. Furthermore a poor prognosis is mentioned, but the prognosis of BL is actually good.
2. The references in the introduction are missing or wrong. Ref 2 is not right for the pathogenesis of BL, ref 15 is better although it is a review. In the paper used it shows that the BL cell lines are all wild type and have no mutations, the upregulation of EZH2 seems only tob e caused by the overexpression of Myc. There is no reference for the presence of the HER2/neu pathway in BL, no real evidence and so the rational seems very weak.
3. The concentrations of the inhibitors are quite high, concentrations under 1 uM would be better.
4. Results: the description of the Xbp1 and Chop/Bip is limited, in the figure there is little effect on Ramos, in the tekst this is not mentioned.
5. The effect on cyclin D1 : a protein expression would be perferred over RT-PCR.
6. Some of the figures have a bad resolution.
7. In figure 3 there is a mistake in C.
8. In figure 5 just the data on apoptosis would be sufficient, and the X-axis should be the same in all figures.
9. P53 in figure 6 should also be western blot.
Reviewer 2 Report
The manuscript investigates possible synergism of MS1943, EZH2 degrader and lapatinib, dual inhibitor of EFGR and HER2 in Burkitt lymphoma cell lines. The authors have shown the potential combination being effective with the clear biological rationale. Only minor revisions are needed. In the introduction section the authors need to expand the part on Burkitt lymphoma including the prognosis and the current standard of treatment followed by the unmet medical need in this entity in order to provide the rationale for the study. Furthermore, EZH2 inhibitors and their efficacy in a variety of neoplasms should be explained in more details.
Round 2
Reviewer 1 Report
The manuscript is much improved